# Fast forward: Rephrasing 3D deformable image registration through density alignment and splatting

**Mattias P. Heinrich**[1] iD                    MATTIAS.HEINRICH@UNI-LUEBECK.DE
[1] *Institute of Medical Informatics, Universität zu Lübeck, Germany*
**Alexander Bigalke**[1,2] iD
[2] *Drägerwerk AG & Co. KGaA, Lübeck, Germany*
**Lasse Hansen**[3] iD                    LASSE@ECHOSCOUT.AI
[3] *EchoScout GmbH, Lübeck, Germany*

**Editors:** Accepted for publication at MIDL 2025

## Abstract

Unsupervised learning- and optimisation-based 3D registration has almost exclusively been approached using backward warping (interpolation) for transforming images. While this has practical advantages in particular the ease of implementation within common libraries it limits the robustness and accuracy in certain challenging scenarios. The alternative solution of forward splatting (extrapolation) is currently limited to very few applications, e.g. mesh or point cloud registration, requiring specific geometric learning architectures that are so far less efficient compared to dense 3D convolutional networks. In this work, we propose to use a straightforward forward splatting technique based on differentiable rasterisation. Contrary to prior work, we rephrase the problem of deformable image registration as a density alignment of rasterised volumes based on intermediate point cloud representations that can be automatically obtained through e.g. geometric vessel filters or surface segmentations. Our experimental validation demonstrates state-of-the-art performance over a wide range of registration tasks including intra- and inter-patient alignment of thorax and abdomen.

**Keywords:** Deformable registration, Splatting, Warping

## 1. Introduction

Deformable image registration is ubiquitously used in various medical imaging applications many of them requiring realtime and fully automatic solutions. Despite significant progress over the last few years, learning-based image registration still faces some challenges that are absent for other tasks, e.g. a larger performance discrepancy of feed forward models with or without instance optimisation or fine-tuning (Balakrishnan et al., 2019; Heinrich and Hansen, 2022; Hering et al., 2022). The gap before and after adaptation to a particular instance can be as large as 50% (Heinrich and Hansen, 2022; Wang et al., 2022). The necessity to use more complex multi-stage (Mok and Chung, 2020) or cascaded (Zhao et al., 2019) architectures to mimic an iterative alignment is well documented for large deformation tasks beyond the cranial vault. Moreover, recent research even explored to explicitly build upon parts of conventional gradient descent methods to stabilise the learning of registration networks (Jia et al., 2021). Eventually, some approaches directly focused on speeding up the optimisation of pre-computed geometric or semantic features (Siebert et al., 2021).

A challenge for **unsupervised deformable image registration** is to find a balance between similarity metric and smoothness regularisation. To avoid degenerated solutions

- that e.g. let to two points from different origins in the moving image correspond to the same location in the fixed scan - a smooth mapping is essential. Commonly, a combination of several regularising elements is employed: 1) constraints on the gradient of the estimated displacements, which can also be a conditional hyperparameter (Mok and Chung, 2021); 2) employing a low-parametric transformation model such as B-splines (Qiu et al., 2021) or Fourier-basis functions (Jia et al., 2023); 3) a symmetric registration framework that achieves inverse consistency by design (Greer et al., 2023). The latter two are preferable because they do not introduce any additional continuous hyperparameters and point to an apparent deficiency of oftentimes weak regularisation constraints. All mentioned approaches formulate registration as a backward warping, which interpolates intermediate intensity values from the moving image at displaced locations.

Contrarily, sparse **point cloud registration** can be formulated as forward transform, which aims to align two 3D densities e.g. by solving an optimal transport problem (Shen et al., 2021) or using the Chamfer distance between moved and fixed points. This, however, comes with its own challenges for defining a well-differentiable cost function (Bigalke and Heinrich, 2023; Heinrich et al., 2023). Point cloud processing has greatly benefited from the advent of new geometric deep learning architectures (Zhao et al., 2021). Yet its inherent limitations of expensive sparse memory access (Liu et al., 2019) have prevented a more widespread adoption. A notable popular point registration networks (Wu et al., 2020) was only cited 150 times to date. In addition, using forward transforms on dense images requires the extrapolation pixel intensities from one grid to another, a so called splatting operation, that may lead to holes that could negatively affect the loss computation.

We hypothesise that revisiting an under-researched paradigm for medical image registration as forward transform model that combines the complementary strengths of dense and sparse model components can alleviate most aforementioned problems. This leads to a substantially different solution, which replaces the spatial alignment of dense images using backward warping with a forward splatting of density estimates of an intermediate point cloud representation. Prior work (Heinrich et al., 2023) already showed that splatting provides a well-differentiable loss for training graph-/point registration networks. We crucially extend upon this work by combining forward splatting with well-proven dense 3D convolutional architectures for deformation estimation. Our new paradigm for defining the loss and transform leads to more robust and accurate performance without finding additional hyperparameter for constraining the displacement regularity.

Our method combines dense volumetric registration with intermediate point cloud representations with the following three **technical contributions**:

- Conversion of volumetric scans into point based multichannel density representations using either semantic, geometric or intensity-based features

- Differentiable rasterisation for an unsupervised loss definition within a multi-step and optionally symmetric, inverse-consistent registration framework

- Fully-convolutional dense 3D networks adapted to rasterised point cloud inputs for improved efficiency and accuracy

To demonstrate the efficacy of this newly proposed paradigm for deformable registration we perform a detailed analysis on a challenging proof-of-concept dataset followed by compre-

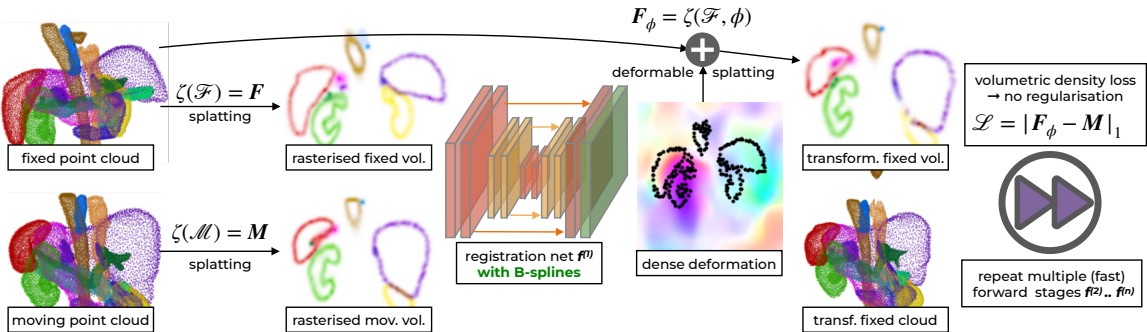

Figure 1: Given two sparse point clouds $\mathcal{F}$ and $\mathcal{M}$ we rasterise 3D volumes $\boldsymbol{F}$ and $\boldsymbol{M}$ by **forward splatting** $\zeta$ as input to a registration U-Net (including a B-spline transform) which predicts a dense deformation $\phi$ that is sampled sparsely at the points of $\mathcal{F}$ (vector addition). An $L_1$ loss of a newly splatted $\boldsymbol{F}_\phi = \zeta(\mathcal{F}, \phi)$ and $\boldsymbol{M}$ is used to derive a well differentiable loss and **stable multi-stage model**.

hensive experimental validation on 3D medical datasets involving large deformation lung registration and cross-patient shape alignment of abdominal organs. The source code is available at https://github.com/mattiaspaul/fastforward.

## 2. Method

We consider the task of aligning two 3D images - fixed $\boldsymbol{F}$ and moving $\boldsymbol{M}$ - with intermediate representations as sparse 3D point cloud, denoted as $\mathcal{F}$ and $\mathcal{M}$ - using a spatial transform $\varphi$. Given unlabelled training data $\mathcal{T} = \{(\boldsymbol{F}_i, \boldsymbol{M}_i)\}_{i=1}^{|\mathcal{T}|}$ comprising $|\mathcal{T}|$ unaligned pairs we aim to learn a function $f$ represented by the parameters $\theta_f$ of a deep neural network to predict a displacement field $\hat{\varphi} = f(\boldsymbol{F}, \boldsymbol{M}, \theta_f)$ that minimises a dissimilarity metric $\mathcal{S}(\boldsymbol{F}, \varphi \circ \boldsymbol{M})$.

**Forward versus backward transforms:** In the above notation a conventional backward transformation was formulated that *pulls* $\boldsymbol{M}$ towards $\boldsymbol{F}$. But it is similarly possible to make the model learn a forward transform $\phi$ that minimises $\mathcal{S}^*(\phi \circ \boldsymbol{F}, \boldsymbol{M})$. Let us start with volumetric data that resides on a regular cartesian grid with integer coordinates $\boldsymbol{q} = (q_x, q_y, q_z) \in \mathbb{N}^3$ and associated voxel intensities $\boldsymbol{x} \in \mathbb{R}^{n_x \times n_y \times n_z}$ (for ease of presentation we limit ourselves to grayscale but extend the concept to multi-channel images later). To apply the spatial transform $\varphi$ to $\boldsymbol{M}$ in a differentiable manner, we can follow (Jaderberg et al., 2015), which introduced spatial transformer networks. In practice, we are interested in the intensity values for the coordinates $\boldsymbol{p}_\varphi = \boldsymbol{p}_0 + \Delta\boldsymbol{p}$, which represent the addition of identity transform and the relative displacements. We follow the notation of (Dai et al., 2017) to define a trilinear interpolant $G(\boldsymbol{q}, \boldsymbol{p}) = g(q_x, p_x) \cdot g(q_y, p_y) \cdot g(q_z, p_z)$, with $g$ being defined as $g(q_x, p_x) = \max(0, 1 - |q_x - p_x|)$, which can be used to perform a weighted average over the 8 neighbouring grid points of displaced voxels $\boldsymbol{p}_\varphi$ using:

$$\boldsymbol{y}(\varphi, \boldsymbol{x}) = \sum_{\boldsymbol{q}} G(\boldsymbol{q}, \boldsymbol{p}_\varphi) \cdot \boldsymbol{x}(\boldsymbol{q}). \tag{1}$$

Intuitively speaking, this backward transform $\boldsymbol{M}_\varphi = \boldsymbol{y}(\varphi, \boldsymbol{M})$ gathers information in an irregular manner from the moving image onto a regular grid for the new warped image that should reside in the domain of the fixed image. Each grid point in the warped image receives the same density, yet the voxels from the moving image may contribute differently to the transformed output. A forward transform conversely scatters voxel intensities from a regular grid onto irregularly spaced spatial coordinates in the transformed image, which requires an extrapolation to neighbouring grid locations.

Hence as detailed in (Heinrich et al., 2023) for the case of sparse 3D point clouds $\mathcal{F}$ and $\mathcal{M}$, the order of Eq. 1 has to be reversed to $\boldsymbol{x}(\phi, \boldsymbol{q}) = \sum_{\boldsymbol{p}} G(\boldsymbol{p}, \boldsymbol{q}_\phi) \cdot \boldsymbol{y}(\boldsymbol{p})$ resulting in a trilinear **splatting operation** $\boldsymbol{F}_\phi = \boldsymbol{x}(\phi, \boldsymbol{F}) = \zeta(\mathcal{F}, \phi)$. When used in isolation this may lead to visually undesirable effects, e.g. holes in the output and multiple intensities that are accumulated onto the same location. While most prior work considered this a disadvantage that required careful post-processing (cf. (Birkfellner et al., 2005)), we want to highlight the benefits of such an approach. When incorporating the splatting operation into the dissimilarity loss, the optimisable function $f_\theta$ (e.g. a deep neural network) has to predict a $\phi$ that retains a similar density distribution across images to avoid penalties, which helps to prevent unrealistic and implausible deformations. To that end, we focus our empirical evaluation on the gains that may be obtained by replacing backward with forward transformations.

**Dense-to-sparse-to-dense model:** To efficiently handle the challenges of splatting large 3D volumes in a differentiable manner we propose to design a model with intermediate sparse point cloud representations. Fig. 1 starts from 3D scans have been converted into two unordered 3D point clouds $\mathcal{F} \in \mathbb{R}^{N_\mathcal{F} \times 3}, \mathcal{M} \in \mathbb{R}^{N_\mathcal{M} \times 3}$. In case of multi-object (multi-channel) registration, each point can have a one-hot vector representing its class assigned. While geometric deep learning has seen use in the medical domain for shape classification and pose recognition (Bigalke et al., 2023), deformable 3D registration has so far been limited to few sparse point cloud based architectures such as PointPWC-Net (Wu et al., 2020). They additionally pose a particular challenge for unsupervised learning to define suitable loss functions (cf. (Shen et al., 2021)). We hence generate a rasterised 3D density representation using the efficient splatting function $\zeta_\sigma$ proposed in (Heinrich et al., 2023) to transform $\mathcal{F}$ and $\mathcal{M}$ into $\boldsymbol{F}$ and $\boldsymbol{M}$, which reside on a regular cartesian 3D grid and where $\sigma$ defines a spatial smoothing kernel. Such an intermediate dense representations was used e.g. in V2V-PoseNet (Moon et al., 2018) to address slow random memory access in k-nearest neighbour graph networks (cf. (Liu et al., 2019)). With 3D voxel volumes at hand we can employ a straightforward 3D U-Net (Çiçek et al., 2016) to define our neural network function $f$. For robustness, the output of the U-Net $\varphi = f(\zeta_\sigma(\mathcal{F}), \zeta_\sigma(\mathcal{M}))$ is fed into a differentiable B-spline transformation (that comprises no trainable parameters) akin to (Qiu et al., 2021). To extract the sparse motion vectors that align the two point clouds we first compute a trilinear interpolation (see Eq. 1) of $\varphi$ at the coordinates of $\mathcal{F}$ to obtain $\phi = \boldsymbol{y}(\mathcal{F}, \varphi)$. Next, we perform another *deformable* splatting to perform a forward transform that should satisfy the similarity $\zeta_\sigma(\mathcal{F} + \phi) \sim \zeta_\sigma(\mathcal{M})$. Note, that absolute and relative coordinates are simply added for the transformation. The hyperparameter $\sigma$ together with the resolution of the cartesian grid $\boldsymbol{q}$ defines how much spatial overlap neighbouring points have within the 3D density estimation of $\zeta$ and improves differentiability.

**Multi-step, regularisation and inverse consistency:** When designing a backward warping baseline, we noted that the implicit regularisation of the B-spline transform is sometimes not sufficient to achieve competitive performance due to the tendency of this model to *pull* points from multiple different locations to the same grid coordinate. The baseline model hence adds a diffusion regulariser $\mathcal{R} = ||\lambda \cdot \nabla \varphi||^2$ to penalise large gradients of $\varphi$, which introduces another hyperparameter to be tuned (cf. (Mok and Chung, 2021)). To capture large deformations, we consider the multi-step approach of (Mok and Chung, 2020) that cascades (Zhao et al., 2019) multiple U-Nets to obtain $\varphi^*_{\boldsymbol{FM}} = \varphi^1 \circ \varphi^2 \circ \ldots \circ \varphi^{(n)}$. Furthermore, we explore the very recent strategy of establishing symmetry and inverse consistency of the registration by construction following (Greer et al., 2023) and estimate $\varphi^* = \exp(f_{\boldsymbol{FM}} - f_{\boldsymbol{MF}})$, where exp represents the exponentiation step of the scaling-and-squaring approach (Avants et al., 2008) (and the output of $f_{\boldsymbol{FM}} = f(\boldsymbol{F}, \boldsymbol{M})$ is scaled down appropriately). When estimating large deformations a single network is usually unable to capture the complex transformation. Hence, a two-step consistent approach with two networks $f^{(1)}$ and $f^{(2)}$ is used:

$$\varphi^{(2)} = \exp\left(\frac{f^{(1)}_{\boldsymbol{FM}} - f^{(1)}_{\boldsymbol{MF}}}{2}\right) \circ \exp(f^{(2)}_{\hat{\boldsymbol{F}}\hat{\boldsymbol{M}}} - f^{(2)}_{\hat{\boldsymbol{M}}\hat{\boldsymbol{F}}}) \circ \exp\left(\frac{f^{(1)}_{\boldsymbol{FM}} - f^{(1)}_{\boldsymbol{MF}}}{2}\right). \tag{2}$$

Here, two mid-way warps are defined as $\hat{\boldsymbol{M}} = \boldsymbol{M} \circ \exp(\frac{f^{(1)}_{\boldsymbol{MF}} - f^{(1)}_{\boldsymbol{FM}}}{2})$, similar to the ANTs SyN approach (Avants et al., 2008). More details on the implementation of each step can be found within our open source code.

## 3. Experiments

We consider two tasks that are currently challenging for learning based registration and often require additional instance optimisation. Our aim is to ease the difficulty by replacing backward warping with forward splatting in two image registration tasks. First, the large deformation 3D CT lung registration between inspiration and expiration using the PVT1010 (Shen et al., 2021) and DIRlab COPD (Castillo et al., 2013) patient data respectively. And second, finding shapes correspondences across patients of the multi-organ abdominal CT dataset Beyond the Cranial Vault (BVC) with significant differences across anatomies (Xu et al., 2016) that was used in Learn2Reg 2020 (Hering et al., 2022). We perform up to three registration approaches for those datasets: 1) purely optimisation based, 2) asymmetric (multi-stage) deep learning and 3) symmetric two-step inverse-consistent deep learning registration.

The DIRlab COPD dataset remains one of the most challenging tasks for deep learning registration yielding target registration errors (TRE) of $> 7$ mm for VoxelMorph (Balakrishnan et al., 2019) and $\approx 5$ mm for LapIRN (Mok and Chung, 2020). The only highly accurate solution we found in the literature (Greer et al., 2023) that achieved $< 2$ mm was trained on additional 1000 pairs of high-resolution intensity scan pairs of the COPDgene study. The related point cloud dataset (PVT1010) has been tackled with the sparse Point-PWC (Wu et al., 2020) model for which results of $\approx 5$ mm before instance optimisation (Shen et al., 2021; Heinrich et al., 2023) were reached when trained with unsupervised losses and 2.3 mm for supervision with simulated deformations followed by mean teacher domain adaptation on the real unlabelled data.

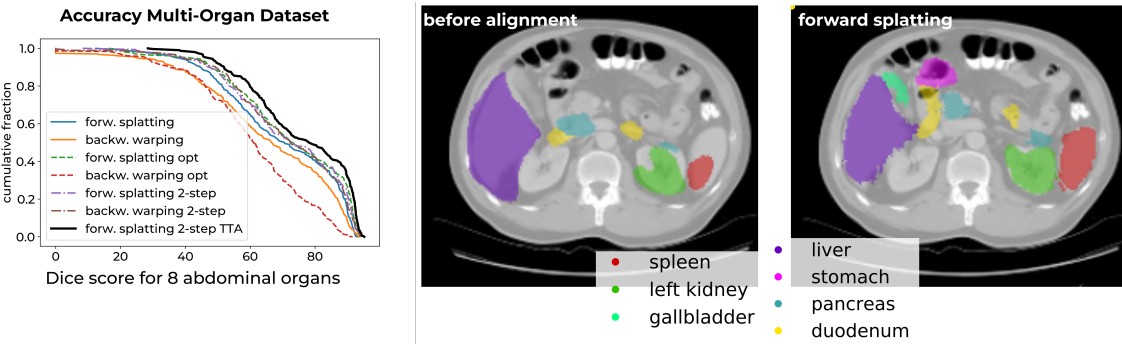

Figure 2: Example distribution and case for the abdomen test data with initial and trans-formed labels. A substantial improvement can be found using forward splatting.

**PVT1010 / COPD** comprises point clouds within the paired lungs of 1000 patients that depict $\approx$ 50-120$\times 10^3$ well distributed 3D coordinates at lung vessels, airways or other points of high local image gradient or curvature in both extreme respiratory phases. For 10 scans the publicly available DIRlab cases provide additional image information (which we do not use) and serve as a benchmark with 300 manual landmarks each.

The **multi-organ abdomen** dataset (AbdomenCT) is made up by 30 CT scans of different patients for pairwise training (Xu et al., 2016), whereas the test set has 20 scans with 9 annotation labels (Gibson et al., 2018). To create multilabel point-clouds we employ predictions from a pre-trained TotalSegmentator (Wasserthal et al., 2023) for 14 anatomies and sample 61'440 (15x4096) surface points (using both non-maximum suppression and farthest point sampling). We use `torch.gradient` to obtain spatial gradients of the seg-mentations represented as one-hot tensors. The FPS implementation followed the code from (Zhao et al., 2021). The whole process requires $\approx$ 200 millisec. per 3D volume. The Gaussian smoothing employed in the differentiable splatting function $\zeta_\sigma$ is sped-up using the approximation of (Kovesi, 2010) and extended to the 15-channel input.

**Implementation details**: We employ the basic MONAI U-Net (Cardoso et al., 2022) with 5 levels and up to 64 channels as $f$ with 2 or 30 (multi-organ) input channels of the con-catenated rasterised point clouds with fixed dimensions of $128^3$ voxels. The output is fed to a hyperbolic tangent and multiplied by 0.25 to limit the displacement range and regularised using a cubic B-spline with a control-point grid of $32^3$ (see Appendix for hyperparameter choices). In case of the two-step inverse consistency (IC), we use scaling-and-squaring and symmetry by design (see Eq. 2) with two trainable networks. For multi-step, we train up to $st = 8$ U-Nets, further limit each output to $\frac{0.3}{st}$ and compose the output transformations. The dissimilarity loss $\mathcal{S}$ is chosen as the $L_1$-Norm of the splatted point clouds of fixed and transformed moving scan. All models or respectively the B-spline grid for optimisation-only approaches are trained with Adam with a learning rate of 0.015 for 10'000 or 150 iterations respectively. For the latter, the backward warping loss led to insufficient perfor-mance, which we addressed by weighting the $L_1$-Norm with the fixed point cloud density. Test-time adaptation is optionally used to fine-tune the weights of $f^{(1)}$ and $f^{(2)}$, since our model uses only an unsupervised loss on the automatically generated point clouds (with

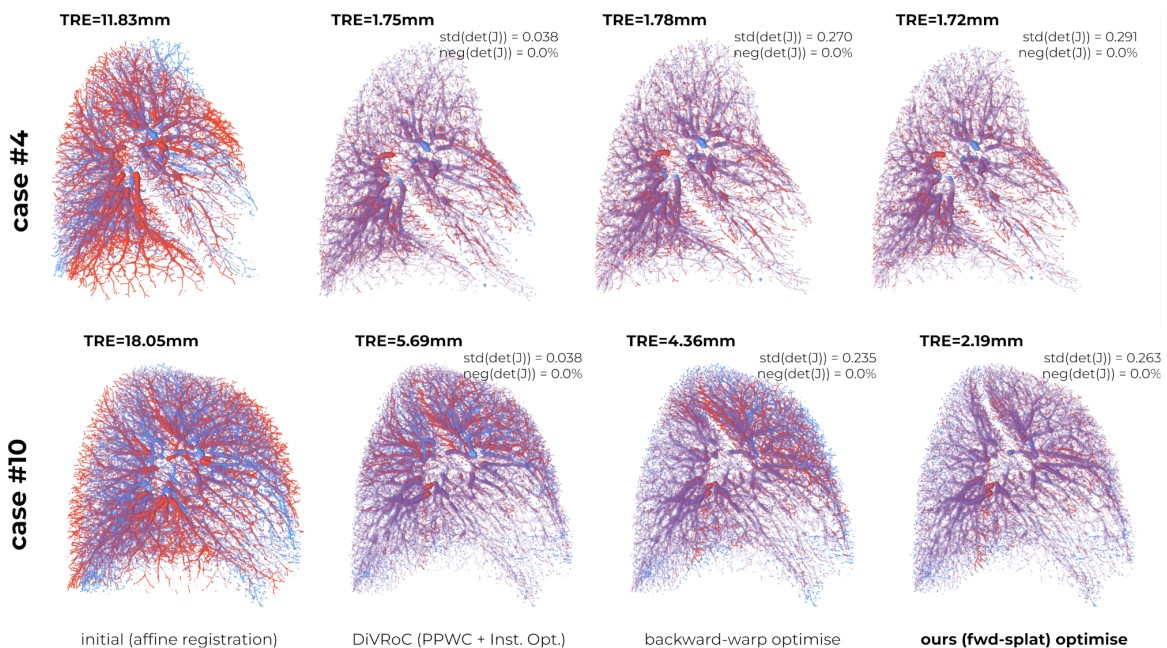

Figure 3: Rendering of inspiration to expiration point clouds (PVT1010/COPD) showing improved alignment (purple) of our method compared to backward warping.

Table 1: Quantitative accuracy for **multi-organ abdomen** averaged across 8 labels (Gibson et al., 2018) (initial Dice of 25%). All methods use label-supervision but only during training. Our forward splatting outperforms backward warping in every case, in particular for optimisation and TTA and sets new SOTA.

| method | optimisation | 8x multi-step | two-step IC | net+TTA |
|---|---|---|---|---|
| backw. warp. | 59.88±7.7% | 64.56±9.1% | 70.95±5.4% | 61.42±6.7% |
| forw. splat. | 71.74±6.3% | 69.10±6.3% | 71.25±5.2% | **75.29±4.0%** |
| SOTA img | 69% | 67% | 52.3% | 69.6% |
| /label | (Siebert et al., 2021) | (Mok and Chung, 2020) | (Heinrich and Hansen, 2022) | (Heinrich and Hansen, 2022) |
| | ConvexAdam | LapIRN | VxM+ | VxM+(+IO) |

$N_{\mathcal{F}} = N_{\mathcal{M}} \approx 64 \cdot 10^3$). This step requires only $3 - 5$ secs. per scan pair for $20 - 50$ iterations. Training requires less than 1 hour per model on one GPU.

This is achieved by the very efficient implementation of the splatting operation to rasterise input and intermediate (warped) point clouds, which resulted in orders of magnitude faster processing compared to traditional sparse data interpolation and modern GPU-accelerated kernel operators (Charlier et al., 2021), see Appendix C for details.

Table 2: Manual landmark error (TRE) in mm for **PTV1010/COPD** (Castillo et al., 2013) with an initial affine TRE of 11 mm. We set new SOTA for point cloud methods and outperform most approaches that use intensity images (values after / denote inst. optimisation)

| method | optimisation | 8x multi-step | two-step IC | net+TTA |
|---|---|---|---|---|
| backw. warp. | 3.21±4.0 | 4.00±2.1 | 3.2±2.4 | 1.82±0.3 |
| forw. splat. | 2.00±0.6 | 2.16±0.7 | 3.1±2.4 | **1.76**±0.4 |
| SOTA | **0.82** | 4.99 | 2.03/1.62 | 2.2 |
| images | (Rühaak et al., 2017) | (Mok and Chung, 2020) | (Greer et al., 2023) | (Heinrich and Hansen, 2022) |
| SOTA point | 3.13(CPD) | | 2.31 | 5.96/2.39 |
| clouds | (Falta et al., 2024) | | (Bigalke and Heinrich, 2023) | (Heinrich et al., 2023) |

## 4. Results and Discussion

Our empirical results demonstrate clear advantages compared to prior work. We outperform all previous point cloud registration models on the challenging PVT1010 dataset by a large margin (see Tab. 2 and Fig. 3), when replacing a sparse graph-convolution based model (Point-PWC) with a dense multi-stage (symmetric) U-Net, while keeping the attractive properties of sparse inputs. In particular, our novel integration of forward splatting within a cascade of dense U-Net registration networks leads to a threefold reduction in TRE compared to DiVRoC (Heinrich et al., 2023) (5.96→2.16 mm) before instance optimisation. Unlike prior methods, we dynamically splat moved points after each transformation rather than relying on backward warping, which significantly improves registration accuracy. Point clouds are efficient for data sharing and substantial reduce the risk of privacy leakage (cf. (Shen et al., 2021)). However, traditional point cloud networks suffer from unstructured memory access and expensive kNN queries, particularly as the number of points increases. In direct comparison to DiVRoC (Heinrich et al., 2023), which also employs rasterisation as loss, our method achieves a sixfold speed-up while reducing computational complexity: the PPWC model used processed only a subset of 8192 points per scan and required eight partial runs to predict a forward transform in ∼1.5 seconds, using 7.7M parameters and 65.2 GFlops. In contrast, our sparse-to-dense representation enables U-Nets that are independent of the number of points, requiring only ≈0.25 seconds for the two-step inverse consistency model, with 1.2M parameters and 47.2 GFlops. This highlights the efficiency gains of our method, making it highly scalable while retaining the expressiveness and privacy advantages of sparse point clouds. By extending the concept to multi-channel surface point clouds, where points carry categorical labels, we outperform all previous supervised image-based methods for AbdomenCT (+2%) and achieve further gains to +5% points when employing test-time adaptation. Notably, the stabilising effect of forward splatting is more pronounced for multi-step or iterative alignment, while the two-step inverse consistency (Greer et al., 2023) introduces complementary benefits, enforcing perfect symmetry in all solutions. A supplementary ablation study (Appendix) empirically shows the sensitiv-

ity of hyperparameter choice for the warping strategy, whereas our method remains robust across a wider range of values. Further improved alignment can still be obtained when considering the complete dense image information (see superior results for (Greer et al., 2023) in Tab. 2), which would motivate hybrid models trained on large-scale, multi-centric, privacy-preserving and modality-agnostic point cloud datasets, with local fine-tuning on full patient scans at test time.

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

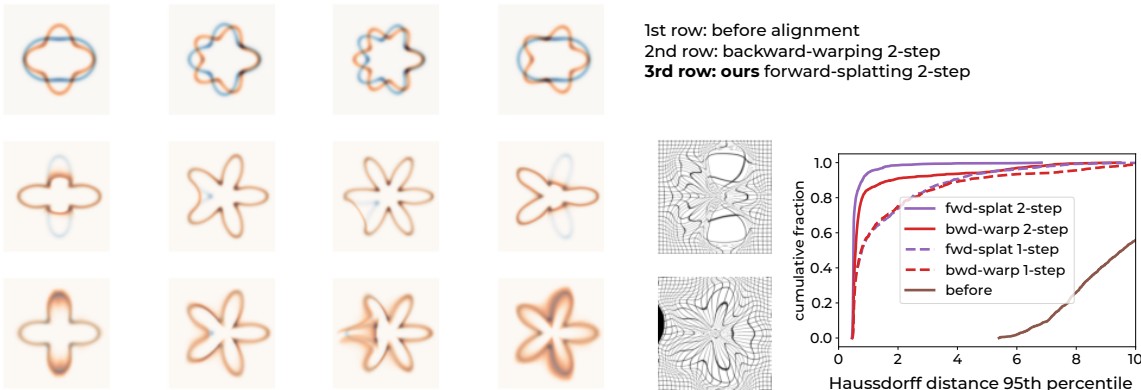

Figure 4: Various cases of our petal dataset with topological changes to be aligned (showing the case with the $5^{th}$ highest error out of 64 samples each). Forward splatting achieves more accurate alignment and lower Haussdorff distances along with more plausible displacement grids.

Shengyu Zhao, Yue Dong, Eric I Chang, Yan Xu, et al. Recursive cascaded networks for unsupervised medical image registration. In *Proc ICCV*, 2019.

## Appendix A. Synthetic petal dataset

We additionally perform experiments on a highly deformable synthetic 2D task with the aim to align flower shapes with different numbers of petals. Petals can be flexible generated by sampling 1024 points along the angle $\alpha$ with the equation $r = \frac{\cos(k \cdot \alpha) + d}{d}$, where $k \in \mathbb{N}$ defines the number of petals and $d \in \mathbb{R}^+$ modulates the curvature of the leaves. With different numbers $k_1$ and $k_2$ of petals in fixed and moving image as well as changes in curvatures, huge local deformations have to be estimated (see Fig. 4). Results show that the warping loss produces unrealistic transforms, which may explain its difficulty to further improve through test-time adaption for the multi-organ data.

## Appendix B. Hyperparameter search

Figure 5 shows exemplary hyperparameter sweeps for the $10^{th}$ COPD case. We perform an 8×8×8 sweep for our approach and evaluate an additional 8 values of $\lambda$ each for backward-warp. An optimal valley can be found for both approaches for the resolution of the rasterisation grid versus the employed B-spline smoothing (equivalent Gaussian $\sigma$), whereas the optimal choice for the smoothing of the extrapolation (splatting $\zeta_\sigma$) is more stable for forward splatting. The comparison of B-spline smoothing and diffusion $\lambda$ reveals the finicky behaviour of backward warping, which deteriorates sharply when moving slightly away from the optimum. While the population based learning and inverse-consistency can alleviate this choice to some degree at highlights the current limitation of many unsupervised registration algorithms.

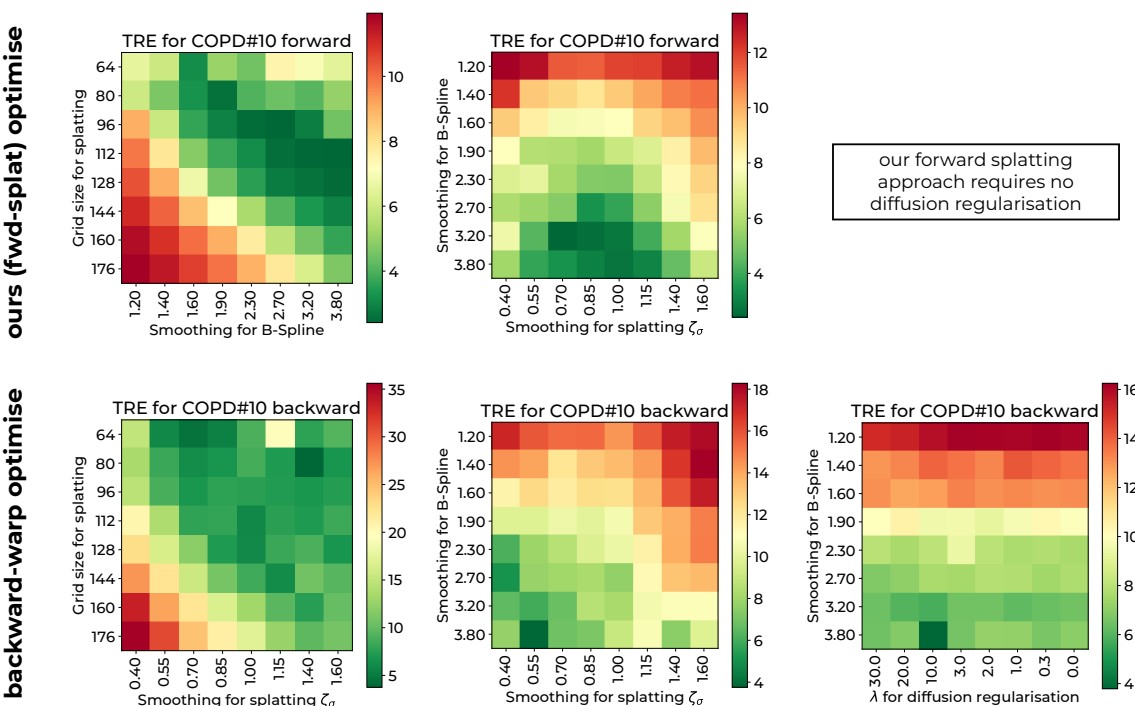

Figure 5: Hyperparameter search for the optimisation based alignment of the $10^{th}$ COPD case comparing our proposed forward splatting with only intrinsic regularisation with conventional backward warping that employs additional diffusion regularisation. Dark green values indicate accurate registrations.

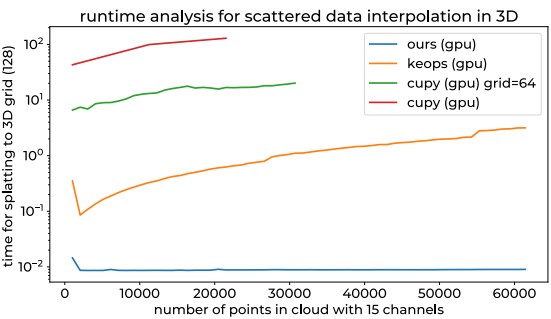 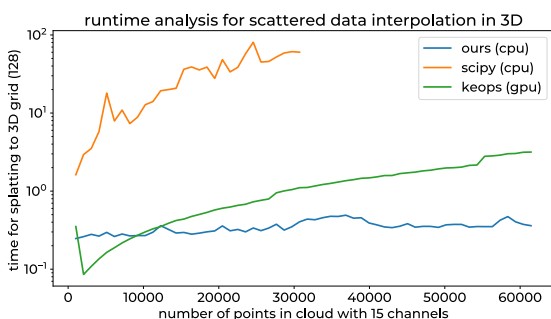

Figure 6: Efficiency analysis that shows the very low runtime of our proposed rasterisation across all point cloud sizes for a dense grid of 128x128x128 with 15 channels. Even on a CPU our approach outperforms the Keops GPU implementation for clouds larger than 12'000 points. Time given in seconds.

## Appendix C. Efficiency of splatting / rasterisation

A detailed comparison of runtimes for our splatting operator that rasterises an input or intermediate (warped) point cloud of varying size to a 3D grid is shown in Fig. 6. It demonstrates a huge speed up of our proposed use of the Jacobian of the `gridsample` operator in pytorch (see also https://github.com/mattiaspaul/fastforward). We benchmarked our approach against several competing implementations: `scipy.interpolate.griddata`, `scipy.spatial.cKDTree` and the highly efficient sparse Gaussian kernel from Keops (Charlier et al., 2021). We demonstrate enormous speed-ups (150-350x fold) and even outperform the most advanced (Keops) GPU implementation on the CPU. Using CUDA our splatting operator rasterises clouds with >60'000 points and 15 intensity channels in <10 millisec. on a 128x128x128 dense grid. Hence there is virtually no overhead compared to the U-Net models.

