# OpenReview forum: "Fast forward: Rephrasing 3D deformable image registration through density alignment and splatting"
_MIDL.io/2025/Conference — MIDL 2025 Poster_

### Official Review · Reviewer_c4p5 · 2025-02-22

**Confidence:** 4
**Preliminary Rating:** 4
**Recommendation:** Poster
**Final Rating:** 5

**Summary:**

The authors propose a Gaussian-splatting-based image registration methodology. They first build a point cloud of the given images using "some form of structural guidance" and then register the rasterized volume. They also explore various regularization strategies, which add depth and interest to the work. Finally, they achieve strong results on abdomen and lung registration datasets.

**Strengths:**

- Significantly improved registration results.
- The method is very clear and well-written.
- Much faster compared to previous methods.
- Authors perform extensive experiments on different regularization methods and their effects.

**Weaknesses:**

- This method always requires segmentation maps or volumetric information of the structures.
- It does not use any intensity-based (image-space) registration losses, which raises a concern: the registration might only align the outlines of the structures, rather than the inner parts of the organs.

**Detailed Comments:**

Please see the questions below.

**Justification Of The Final Rating:**

"The authors propose a well-structured idea with strong evaluation scores. The main drawback of the paper is its reliance on segmentation masks; however, I believe this is acceptable since they explicitly acknowledge it. The overall quality meets MIDL standards, and this is a valuable contribution to the registration community. Therefore, I strongly recommend acceptance.

**Justification Of The Preliminary Rating:**

The paper is well-written, and the methodology is very interesting. The authors achieve strong registration performance. After addressing the reviewers' concerns, I would be happy to consider this paper as a candidate for an oral presentation.

**Questions To Address In The Rebuttal:**

- In the contributions section, the authors state: "Conversion of volumetric scans into point-based multichannel density representations using either semantic, geometric, or intensity-based features." Could you please provide examples of how this can be performed using intensity-based features? What are the common methods for doing this, and how reliable are they? This is important because it could potentially allow the network to be entirely unsupervised.
- How is this method invulnerable to potential segmentation errors that may arise from foundational segmentation models? Do you plan to conduct an ablation study on this?
- Is it fair to claim that it achieves SOTA results compared to completely unsupervised models? Your method is guided by additional structural information. Which comparison methods use segmentation during training? Please clarify and include this information in the tables.
- What are the pros and cons of calculating the similarity loss on both fixed and transformed moving point clouds?
- Is it possible to extend this to multimodal pairs? For example, why weren’t results included for MR-CT registration?
- Assume the given pair of images contains tumors, and the segmentation/volumetric information does not cover the tumor. For instance, there is segmentation for brain structures, but no segmentation for the tumor inside the brain. What would happen in this case? Wouldn't it be a disadvantage not to write the similarity loss function directly on image intensities?

**Special Issue:**

Yes

---

> ### Author Response · Authors · 2025-03-07
>
> We thank the reviewer for their very positive remarks about the strong results, efficiency and well-written clear descriptions in our paper.
>
> Q: Requirement of volumetric point cloud representation without intensity-based losses, which may not inner parts of the organs. Possibility to extend to image features and work without any supervision.
>
> A: We agree with the reviewer that our current focus was on using surface and vessel structures for alignment. We also expect further gains from incorporating additonal salient image features, which is already possible with the extension to multi-channel valued input that could be feature vectors sampled at point cloud locations. We also performed first promising experiments with extracting simpler geometric operators for point extraction, such as local curvature or Canny edges in 3D. Kaftan et al. (https://doi.org/10.1007/s11548-024-03310-z) e.g. trained a lightweight CNN for point extraction for geometric fissure segmentation. Nevertheless even without adding any image-based features our approach already outperformed many competitive image-based baselines (VoxelMorph, LapIRN). This could also be due in part to the avoidance of relying on specific image contrasts.
>
> Q Vulnerability to segmentation errors and potential ablation study. Possbility of multimodal MR-CT registration.
>
> A: These are both great suggestion and should certainly be approached in future work. Given the segmentation models are comparibly accurate for MRI and CT our method directly lends itself to multimodal registration. Segmentation errors are already included in our evaluation (the deformations were estimated on automatic surface labels and evaluted on manual segmentations). But the reviewer correctly points out that the quality of surface label extraction is relevant for the registration accuracy and we will further evaluate this in future.
>
> Q: Clarification on use of segmentations for methods in Tab. 1.
>
> A: All methods use label-supervision but only during training. (we added this information to the caption)
>
> Q: Pros and cons of calculating the similarity loss on both fixed and transformed moving point clouds.
>
> A: Pro: stability and robustness with in general smoother transformations. Con: Difficult or complex deformations, e.g. sliding motion are harder to cpature.
>
> Q: What happens if tumour is within brain structures but not segmented/no surface points?
>
> A: In this case indeed an intensity based registration is preferable.
>
> In addition we added a further efficiency analysis to the Appendix (see comments to other reviewers).

---

> > ### Comment · Reviewer_c4p5 · 2025-03-14
> >
> > I appreciate the authors' detailed discussions and clarifications.
> >
> > I believe this paper makes a valuable contribution to the image registration community and deserves to be included in this conference. Therefore, I am increasing my rating to "accept."
> >
> > Thank you for your good work.

---

### Official Review · Reviewer_VVWb · 2025-02-22

**Confidence:** 3
**Preliminary Rating:** 4

**Summary:**

This paper proposes an approach to unsupervised diffeomorphic deformable registration of 3D volumes, with a key change in strategy, which is to use forward warping (splatting) instead of backward warping. The paper proposes to transform 3D scans into point clouds, which are then rasterised through splatting to give a fixed volume. Deformation fields can be predicted on these rasterised volumes using a fully convolutional network, which is then sampled at points of the original point cloud, followed by a deformation and splatting operation. The authors also add an inverse consistency which guarantees that the predicted deformation field will be invertible. Results on the challenging DIRlab COPD dataset show significant improvements over state-of-the-art methods.

**Strengths:**

* The paper is very well motivated and detailed descriptions of all components provided.
* A method of getting dense deformation grids using rasterised point clouds and forward warping is proposed.
* It is also well evaluated across competing methods and baselines.
* The paper supports the proposes model with a good mathematical basis with regards to warping, invertibility, and the two-step approach.

**Weaknesses:**

* Could the authors clarify how much extra work is added by the point-cloud creation step? Is it something that drastically increases inference time on one pair of volumes, as each volume will need to be converted before being used in the network?
* Backward warping is quite easy to implement in terms of matrix operations. Forward warping is not so straightforward as it requires extra buffers to be created. Could the authors clarify how they implemented fast forward warping to have their models train within 1 hour on GPUs?
* The second paragraph on page 4 says, "When incorporating the splatting operation into the dissimilarity loss, the optimisable function f_\theta (e.g. a deep neural network) has to predict a \phi that retains a similar density distribution across images to avoid penalties, which helps to prevent unrealistic and implausible deformations", but this is not accompanied by a reasoning of why f_\theta would do this. Could the authors give more justification for the claim that the network will predict a similar density distribution to avoid penalties? Figure 1 mentions a volumetric density loss, but the given definition seems to suggest it's a standard L1 loss.
* Similar to the above, is there an implicit way to ensure local relationships between points is preserved after the transform?
* A rather smaller clarification, the post-rasterisation volume in Figure 1 seems to very sparse. It is difficult to imagine how a dense deformation grid could be obtained from this. Could the authors please clarify if the given image is sprase only for the purposes of visualisation or whether these volumes are indeed so sparse?

**Detailed Comments:**

* A small suggestion for equation (2): a better way of formatting the parenthesis around fractions is to use \exp\left( ... \right) instead of \exp( ... ).

**Justification Of The Preliminary Rating:**

This is a very good contribution to deformation 3D volume registration. I am in favour of accepting the paper, yet I am inclined to say "weak accept" at the moment because of some concerns raised above. Nonetheless, I thank the authors for their contribution.

**Questions To Address In The Rebuttal:**

Please see weaknesses

---

> ### Author Response · Authors · 2025-03-07
>
> First, we would like to express our gratitude to the reviewer for their careful assessment and appreciation of the "good mathematical basis with regards to warping, invertibility, and the two-step approach" as well as mentioning that "It is also well evaluated across competing methods and baselines".
>
> Q: Clarify how much extra work is added by the point-cloud creation step?
>
> A: For the PVT1010 dataset the point-clouds were already provided. For the multi-organ abdomen dataset we extended our description as follows: "We use ``torch.gradient`` to obtain spatial gradients of the segmentations represented as one-hot tensors. The FPS implementation followed the code from \cite{zhao2021point}. The whole process requires ≈200ms per 3D volume.". This does not include the time for running the TotalSegmentator. In future work we will explore more concepts for surface extraction - initial experiments with canny edge detection provided good results.
>
> Q: Forward warping is not so straightforward as it requires extra buffers to be created. Could the authors clarify how they implemented fast forward warping to have their models train within 1 hour on GPUs?
>
> A: This is an excellent point, as our highly efficient implementation that uses the Jacobian of the grid_sample operator is key to achieve good performance for forward warping. To demonstrate the advantages we now benchmarked our approach against several competing implementations: ``scipy.interpolate.griddata``, ``scipy.spatial.cKDTree`` and the highly efficient sparse Gaussian kernel from Keops  (Charlier et al., 2021). We demonstrate enormous speed-ups (150-350x fold) and even outperform the most advanced (Keops) GPU implementation on the CPU. Using CUDA our splatting operator rasterises clouds with >60'000 points and 15 intensity channels in <10 millisecs on a 128x128x128 dense grid. Hence there is virtually no overhead compared to the U-Net models. This has now been added with graphs to the appendix and been briefly mentioned at the end of the results section.
>
> Q: Why is it assumed that the network intrinsically predicts simlarly distributed densities to avoid penalties?
>
> A: We thank the reviewer for the interesting remark and believe the emperical evidence in Tab. 2 for "8x multi-step" vs "two-step IC" for the forward and backward model best explains our assumption. Given that backward works reasonabily well when enforcing two-step inverse consistency but forward outperforms it by 46% (2.16 vs 4.00mm TRE) gives a good indication that the density is succesfully carried along multiple warping steps using our model. The density loss stems from the combination of the trilinear extrapolation (rasterisation) with subsequent smoothing followed by an L1 loss, which is different to taking the Euclidean (shortest) distance between point clouds as it incorporates partial overlap within the neighbourhood of points.
>
> Q: Clarification regarding sparsity of rasterisation in Fig. 1.
>
> A: This is indeed a difficulty in visualisation as we aimed to combine colour overlays together with the density estimate but only used a single slice of the rasterised volume. Fig. 3 that shows high-resolution renderings of lung point clouds better demonstrates the rich information content of the available clouds with tens of  thousands of points.

---

> > ### Comment · Reviewer_VVWb · 2025-03-13
> >
> > I thank the authors for their answers to questions of all the reviewers.
> >
> > Regarding my point about the loss encouraging preserving of density, if I understand your explanation correctly, in a backward warping setting, the loss is computed only between individual pairs of voxels, as each output voxel gets its information from some input voxel; while in the forward warping setting, the splatting operation would ensure that each point would contribute a certain value to a neighbourhood of voxels at its output location, and thus an output voxel gets weighted information from several points and this preserves density. Would that be correct? If yes, could we have the same case in backward warping, where each output voxel gets weighted information from a neighbourhood of input voxels, in case the coordinates of a point are non-integer? In any case, I encourage the authors to clarify this point in their text. If I have understood correctly, then forward warping also seems important to this density-preserving aspect of the proposed framework.
> >
> > Regarding my point about the implementation of forward warping, this could be an excellent contribution of the paper. I encourage the authors, if possible, to release the implementation of their forward warping algorithm (please ignore in case it is already released and I have missed it). Maybe you could also add some text on its implementation in Appendix C.
> > By the way, the Figure 6 referenced in Appendix C is missing units on the y-axis of both the graphs.

---

> > > ### Author Response · Authors · 2025-03-13
> > >
> > > Q   in a backward warping setting, the loss is computed only between individual pairs of voxels, .. while in the forward warping setting, the splatting operation would ensure that each point would contribute a certain value to a neighbourhood of voxels at its output location, ... Would that be correct?
> > >
> > > A Yes this is the correct assumption. If multiple points are drawn to the same voxel in backward warping, their values=density is averaged and not preserved, whereas forward splatting would add multiple points that are moved to the same location - enabling density preservation.
> > >
> > >
> > >
> > > Q If yes, could we have the same case in backward warping, where each output voxel gets weighted information from a neighbourhood of input voxels, in case the coordinates of a point are non-integer? .., then forward warping also seems important to this density-preserving aspect of the proposed framework.
> > >
> > > A Indeed we had already attempted to improve backward warping following this suggestion, see Implementation details at the end of page 6 "For the latter, the backward warping loss led to insufficient performance, which we addressed by weighting the L1-Norm with the fixed point cloud density." We will strengthen this point to further clarify our reasoning and contribution in the final version as suggested.
> > >
> > >
> > > Q Regarding my point about the implementation of forward warping, this could be an excellent contribution of the paper. I encourage the authors, if possible, to release the implementation of their forward warping algorithm ..
> > >
> > > A We thank the reviewer for their positive remarks. The code/implementation of the forward warping step was already released with the submission at https://github.com/mattiaspaul/fastforward/blob/main/util_midl25.py - as suggested we will also add some further explanation about this in Appendix C.  We thank the reviewer for pointing our the missing units in Fig. 6 y-axis, the values are given in seconds - this will be fixed.

---

### Official Review · Reviewer_EzwQ · 2025-02-25

**Confidence:** 4
**Preliminary Rating:** 1
**Recommendation:** Poster

**Summary:**

This paper reports a method to perform 3D non rigid registration between airways and organs from CT. The authors proposed to convert the 3D CT volume into point cloud, then optimising a deformation field modelled with 3D B-spline to enforce smooth deformation. The main dataset comprises 1000 CT scans, while testing is done on 10 scans with manual landmarks.  Error is computed using L1.

**Strengths:**

Results are very good improving on existing methods. They are convincing given the size of the dataset. Figure 3 is of high quality, and the writing of the paper is mostly clear.

It seems that the actual contribution is “Fully-convolutional dense 3D networks adapted to rasterised point cloud inputs for improved efficiency and accuracy”, but this needs better explanation. The authors mentioned the computational cost of kNN, is that to search for the closest point to compute the loss? I am guessing that the point are splatted back on the 3D grid using Gaussian so that the 3D Unet can be run on the 3D grid, is that correct?

**Weaknesses:**

The justification of the approach is based on arguing the disadvantages of using forward transform compared to backward transform when performing registration between two images. The authors then argue that forward is better, but cannot be done without issues on a 3D grid.

I am puzzled by this part (~first 3 pages) and struggle to make sense of it. The argument about backward and forward registration has to do with interpolation. Interpolating the backward guarantees that all the points of the grid have intensity to compute an intensity based loss, which is not the case of the forward interpolation.  This is correctly explained by the authors albeit in a convoluted way, but is a well known problem that has been in all textbooks for decades. Casting the paper for 3 pages to solve this particular problem is very strange indeed. To avoid asymmetry, all text books also describe cyclic loss when images are swapped and two losses are computed (or the transformation and its inverse are composed), which is also what the authors do. It is unclear thus why the authors claimed “Furthermore, we explore the very recent strategy of establishing symmetry and inverse consistency of the registration by construction following (Greer et al., 2023) […]”
What this paper proposes is to register the two images without interpolation. Instead they first extract the points of the arterial tree and the organs surface (?) and compute a L1 loss from the density resulting in splatting the points back to the volume. They do not explain how the point are created (are they present in the datasets ?).

Using a 3D B spline to regularized the deformation is not new. There are versions of non rigid ICP, versions of mesh based registration using diffeomorphic smooth deformation. The state of the art for smooth deformation is to use a velocity field that is then integrated to produce a diffeomorphic transformation (e.g. for brain alignment). There is a vast literature on those that the authors do not mention as relevant (e.g. Avants et al, first reference).

Figure 1 show the surface of organs whereas Fig 3 is about airways/vessel trees. There is no explanation of how the images are “rastered” or how the points are extracted.

**Detailed Comments:**

I am concerned that there are  too many unclear statements in this manuscript in its present form to be published as is, or with minor modifications allowed during the rebuttal. See my comments in the weakness section.

Is the splatting done for each iteration of an optimisation process, or is it done only at the beginning? How is the training done?

The paper would be better cast in point 3D volumetric non rigid registration without interpolation, without the need to explain forward backward interpolation. But then more explanation would be required about how the point extraction is done, and how is this different from edge matching with 3D volume registration.

**Justification Of The Preliminary Rating:**

This paper presents a somewhat standard method (point could matching using a smooth spline deformation) as a novel way to solve the very old problem of backward/forward interpolation (sold as a new paradigm). The results are good, but any contribution (matching point density?) is buried in convoluted explanations of well known methods and problems.

**Questions To Address In The Rebuttal:**

How was the point extraction performed?

How does this compared to the many diffeomorphic non rigid registration framework?

Please see the weakness section of my comments.

The contribution lists “Fully-convolutional dense 3D networks adapted to rasterised point cloud inputs for improved e!ciency and accuracy”. The authors need to explain this part better. If the point are splatted on the 3D grid, does the model try to match some point density distribution? If that is the case, how does it compare to 3D volume registration based on smoothed edge detection?

**Special Issue:**

No

---

> ### Author Response · Authors · 2025-03-07
>
> Q: Difficulties of forward interpolaton with previous approaches are correctly explained but well-known.
>
> A: We thank the reviewer for their analysis of our approach and agree with them that the problem of forward transforms has been discusses in many articles over the last decades. We would like to point out that those are primarily focussed on dense image registration without deep learning.
> Hence our method provides a new perspective that provides in our opinion value to the MIDL community both from a theoretical view point but also based on the strong experimental results.
> To better convey this point we changed the key statement to "We hypothesise that ~~establishing a new paradigm~~  **revisiting an under-researched paradigm** for medical image registration as forward transform model that combines the complementary strengths of dense and sparse model components can alleviate most aforementioned problems."
>
>
> Q: Avoiding cyclic contraints or losses has been previously proposed.
>
> A: We appreciate the statement that cyclic losses or inverse consistency were previously thought key to alliveate the issues. Our method, however, demonstrates that the effective use of splatting yielding a density based loss achieves strong performance even without strict inverse consistency. As shown in Tab. 2 the 8x multi-step (asymmetric) networks reach a TRE of 4.00 mm for the conventional backward warping but are improved to 2.16 mm using the proposed forward warping.
>
>
> Q: Relations to non-rigid ICP or other dense diffeomorphic smooth deformation models.
>
> A: We would like to state that their was indeed a detailed state-of-the-art comparison to classic point-based registration - i.e. Coherent Point Drift (CPD, which is usally considered superior to ICP) that only achieved a TRE of 3.13 mm (compared to our best  results with 1.76 mm) on PVT.
>
>
> Q: Figure 1 show the surface of organs whereas Fig 3 is about airways/vessel trees. There is no explanation of how the images are “rastered” or how the points are extracted.
>
> A: As the presented method is versatile to be used with different approaches of point cloud extraction, we use two different variants in the abdomen (surface segmentations) and lung (geometric vessel filters) experiments. As detailed in Sec. 3 Experiments, the PVT1010 / COPD dataset already provides point clouds from the authors of the dataset and the extraction is explained in much detail in Shen, Feydy, et al "Accurate point cloud registration with robust optimal transport." NeurIPS 2021. For the abdominal surface point clouds we briefly described the following in our submission "To create multilabel point-clouds we employ pre- dictions from a pre-trained TotalSegmentator (Wasserthal et al., 2023) for 14 anatomies and sample 61'440 (15x4096) surface points and background (using both non-maximum suppression and farthest point sampling, FPS)." We now further extend this description as follows: "We use ``torch.gradient`` to obtain spatial gradients of the segmentations represented as one-hot tensors. The FPS implementation followed the code from \cite{zhao2021point}. The whole process requires ≈200ms per 3D volume."
>
>
> Q: Concerns about unclear statements.
>
> A: We hope to have clarified some aspects and would like to point out that both other reviewers mentioned the paper was clear, well-written and the contribution was convincing. We hope that the reviewer can change their initial negative view on the paper when acknowledging that we never claimed that forward transforms have not been thought of before. They have indeed been considered in the past and most researchers largely stopped working on them and their use in deep learning based image registration is at most minimal today. Showing strong emperical performance of them is in our opinion a contribution on its own.
>
>
> Q: Is the splatting done for each iteration of an optimisation process, or is it done only at the beginning? How is the training done?
>
> A: Regarding the iterative nature of the splatting (rasterisation) operator, we confirm that indeed this is repeated for each warping step and has to be run online and differentiably to train the registration networks effectively. This avoids the re-interpolation of rasterised images and enables us to use vector addition (see Method section) of multiple relative coordinate updates.
> In the revised version we will also include an runtime analysis with respect to grid-size and number of points in clouds for the following methods:  ``scipy.interpolate.griddata``, ``scipy.spatial.cKDTree`` and the highly efficient sparse Gaussian kernel from Keops  (Charlier et al., 2021). The results show orders of magnitudes of efficiency improvements and is now evaluated in Appendix C.

---

> > ### Comment · Reviewer_EzwQ · 2025-03-11
> >
> > Reading your answer and the other reviews I feel that I am missing something about this paper. I don’t understand why the argument about forward and backward is relevant.
> >
> > I think that the problem you are trying to solve is to warp volumes by matching spatially sparse information (i.e. vessel tree or organ surface). Because of sparsity, volumetric methods do not apply. Your approach is to create a density volume by splatting Gaussian from the information available (the points of the tree/mesh). A similar approach could be to create an edge density map (e.g. some normalization of a gradient). Then you perform a non-rigid spline based non rigid registration on the denser volume (compared to the original point cloud). I don’t see what the forward/backward argument has got to do with this.
> >
> > Couldn’t you have done the same thing using standard backward interpolation and a cyclic loss?
> >
> > Do you mean that you avoid interpolation because you splat at each iteration, therefore re-creating a dense volume every time?

---

> > > ### Author Response · Authors · 2025-03-12
> > >
> > > We thank the reviewer for considering our response and confirm that indeed different methods - apart from our proposed differentiable splatting followed by Gaussian smoothing - are possible to create denser volumetric representations. The reviewer then correctly points out that one can perform a non-rigid spline based registration on this denser volume.
> > >
> > > We would like to clarify that this is exactly what we do as a baseline and refer to as "backward warping". Hence the relevance of the forward/backward argument can be answered with the experimental results in Tab.1 and 2 when comparing "8x multi-step" forward vs backward. Here we see a large improvement for forward from 4.00mm to 2.16mm TRE and 64.56% to 69.10% Dice. We agree that it will be helpful to re-iterate this point in the experimental section.
> > >
> > > As the reviewer suspects the backward interpolation strategy somewhat recovers accuracy when employing a cyclic/symmetric loss as demonstrated in the "two-step IC" column - narrowing the gain for the proposed forward strategy. However, after test-time adaptation the difference are still substantial.
> > >
> > > We would also like to point to the fact that the backward warping approach is much more sensitive to hyperparameter choices as shown in Fig. 5 in the appendix and requires an additional diffusion regulariser (which we do not need for forward splatting). This indicates that the creation of suitable density maps for backward warping is not trivial and we already made substantial efforts to have the strongest baseline possible - that already substantially outperforms most SOTA works including LapIRN and VoxelMorph.
> > >
> > > To summarise, the goal of this paper is to shed light on the understudied topic of forward warping in medical (deep learning based) image registration, and the SOTA results on several datasets versus backward warping (including cycle consistency) show, from our point of view, that the strong prevailing focus on backward warping should maybe be reconsidered and at least discussed in the context of deep learning registration, for which we see an ideal opportunity at MIDL

---

> > > > ### Comment · Reviewer_EzwQ · 2025-03-12
> > > > **I get it**
> > > >
> > > > Thank you for clarifying, I think that I finally get it.
> > > >
> > > > I do think that you should introduce your paper at the beginning along the lines of my comment above, before starting the argument between backward and forward. The story could be something like:
> > > >
> > > > "this paper tries to warp volumes by matching spatially sparse information (i.e. vessel tree or organ surface). Because of sparsity, volumetric methods do not apply. The approach is to create a density volume by splatting Gaussian from the information available (the points of the tree/mesh). Using GS outperform edge density map (e.g. some normalization of a gradient) {is that true?}. Then a non-rigid spline based non rigid registration is performed on the denser volume (compared to the original point cloud). Using forward mapping improves further the performance."

---

> > > > > ### Author Response · Authors · 2025-03-15
> > > > >
> > > > > Since the end of the discussion phase is shortly approaching, we would kindly like to enquire whether the reviewer could consider to reflect the discussion and their now more positive opinion on our paper in their final rating. Thank you.

---

### Author Rebuttal · Authors · 2025-03-07

**Rebuttal:**

revised manuscript for rebuttal

**Supporting Material:**

/attachment/247e9b05ab6f2d3ec0880656fa205d0ea5e91c33.pdf

---

### Meta-Review · Area_Chair_tNUv · 2025-03-22

**Recommendation:** Accept (Poster)
**Confidence:** 4

**Metareview:**

Although originally there were a few concerns from the reviewers, the rebuttal and discussion enabled the authors to get into details and explain some of the confusion and also revise the manuscript. All of the reviewers agreed to accept the paper after the rebuttal, including one reviewer that started off with a very negative review. I agree with this decision.  Overall, this is a great outcome for the discussion phase which enabled scholastic conversation. The paper should definitely be discussed at the conference.